# “Holding in Anger” as a Mediator in the Relationship between Attachment Orientations and Borderline Personality Features

**DOI:** 10.3390/brainsci13060878

**Published:** 2023-05-29

**Authors:** Irene Messina, Pietro Spataro, Sara Sorella, Alessandro Grecucci

**Affiliations:** 1Department of Economics, Mercatorum University, Piazza Mattei 10, 00186 Rome, Italy; pietro.spataro@unimercatorum.it; 2Department of Psychology and Cognitive Sciences, DipSCo, University of Trento and Centre for Medical Sciences, University of Trento, Bettini, 84, 38068 Rovereto, Italy; sara.sorella@unitn.it (S.S.); alessandro.grecucci@unitn.it (A.G.)

**Keywords:** borderline personality, anger, emotion regulation, attachment, personality disorder, suppression

## Abstract

Insecure attachment and difficulties in regulating anger have both been put forward as possible explanations for emotional dysfunction in borderline personality (BP). This study aimed to test a model according to which the influence of attachment on BP features in a subclinical population is mediated by anger regulation. In a sample of 302 participants, BP features were assessed with the Borderline features scale of the Personality Assessment Inventory (PAI-BOR), attachment was measured with the Experiences in Close Relationships-12 (ECR-12), and trait anger and anger regulation were assessed with the State and Trait Anger Expression Inventory-2 (STAXI-2). The results indicated that anger suppression emerged as a significant mediator of the associations between both anxious and avoidant attachment and BP traits, while anger control resulted as a marginal mediator in the association between attachment avoidance and BP. Suppressing anger may reflect different forms of cognitive or behavioural avoidance of anger, which may differ on the basis of attachment orientations. We argue that these results may have important clinical implications: the promotion of anger regulation in BP should be considered a critical treatment goal.

## 1. Introduction

Proneness to emotion dysregulation and insecure attachment are maladaptive features of borderline personality (BP), which can be found to be widely distributed within the population, as well as a psychopathological severity continuum that includes the clinical entity of borderline personality disorder at the most extreme pole [1,2,3,4,5]). Despite the different conceptualizations of BP, which vary in the emphasis they attribute to the role of emotion dysregulation [6,7] or attachment/mentalization disturbances [8], such factors can be viewed as closely and inevitably connected. Nevertheless, researchers have often investigated the roles of emotion dysregulation and attachment disturbance in BP in separate studies; thus, the question of how emotional dysregulation and attachment entwine in this disorder has remained unanswered.

The Diagnostic and Statistical Manual of mental disorders (DSM-IV-TR) describes emotion dysregulation of BP disorder in terms of “affective instability due to a marked reactivity of mood”, with a specific emphasis on anger regulation in terms of “inappropriate, intense anger or difficulty controlling anger” [9]. Studies on emotion dysregulation in BP disorder have shown the existence of general difficulties in regulating emotions [3,10,11] corroborated by neuroimaging evidence of impairments in the fronto-limbic brain circuits implicated in the cognitive control of emotions in BP patients [4,5,12]. The perpetuation of emotional dysregulation in BP can be attributed to the limited use of adaptive emotion regulation strategies and the habitual use of maladaptive strategies (for a meta-analysis see [13]). For example, to regulate emotions, BP patients report relying on suppression [14,15], rumination [16,17,18], experiential avoidance [19,20], and emotional avoidance [21,22]. In the specific case of anger, previous studies have reported high levels of trait anger in BP patients [3,23,24] Furthermore, anger suppression and rumination seem to be prominent factors in association with BP features [3,25,26,27] and have been linked to aggressive behaviours [28,29] and self-harming [30].

Regarding interpersonal difficulties in BP disorder, the DSM [9] describes them as “frantic efforts to avoid real or imagined abandonment” and “unstable and intense interpersonal relationships characterized by alternating between extremes of idealization and devaluation”, which strongly imply the prominence of attachment difficulties. Previous studies have reported that individuals with borderline traits tend to have a perception of others as malevolent [3,31,32] and to view themselves as unlovable and inherently bad [33,34,35,36]. These difficulties, seen through the lens of attachment theory, are believed to derive from maladaptive mental representations of the self and others [37,38]. In agreement, studies on adult attachment clearly converge in indicating a strong association between BP and insecure attachment, together with its inverse relationship with secure attachment [38,39]. Among insecure attachment categories, preoccupied and unresolved–disorganized subtypes emerged as overrepresented in BP disorder in interview-based studies [40,41,42,43] Self-report studies, which have investigated the dimensions of anxiety and avoidance in the adult romantic attachments of BP patients, indicate the dimension of attachment anxiety as the one most strongly correlated with BP traits, even if avoidance was also reported as an associated factor [44].

The connection between attachment and emotion regulation has been emphasized in recent contributions in the field of attachment research [45,46,47]. From this perspective, in cases of distress, individuals with avoidant attachment tend to inhibit or block the activation of the attachment system, and to keep attachment needs and tendencies deactivated, leading to the inhibition or suppression of emotional [48]. On the contrary, individuals with anxious attachment tend to hyper-activate the attachment system, resulting in the chronic intensification of negative emotions that demand attention and care or that emphasize a person’s vulnerability and neediness [49]. An alternative view asserts that the association between attachment and emotion regulation styles may differ depending on the specific emotion involved and its role in attachment-related interpersonal dynamics [50,51]. Indeed, while the outward expression of some negative emotions (e.g., sadness, anxiety, fear, and shame) may serve to elicit attention and maintain proximity with others, the outward expression of anger may potentially reduce the likelihood of others offering support and therefore compromise the maintenance of interpersonal relationships. Due to the scarcity of studies specifically focused on anger regulation, such alternative hypotheses have not been empirically verified [52].

Regarding the interplay between anger regulation and attachment in relation to BP, the available evidence suggests that temperamental variability in anger experience mediates the association between attachment and BP [53,54,55]. Only a few studies have investigated the role of anger regulation as a mediator of this association [56,57,58], suggesting that secure attachment may function as a buffer against BP disorder by enhancing the use of positive emotion regulation strategies, while negative emotion regulation strategies seem to dilute the protective effect of secure attachment [56]. However, to our knowledge, no study has yet investigated how the regulation of anger may influence the association between attachment and BP. Due to the relevance of anger dysregulation in BPD, the investigation of its role as a possible mediator between attachment and BPD features deserves special attention.

In the present study, we aimed to explore the relationship between attachment orientations, BP traits and anger regulation, with a further aim of investigating how anger regulation works in concert with or in opposition to attachment variables, culminating in BP features. To better explore these associations, we considered several domains of BP features (affective instability, identity problems, negative relationships, and self-harm). Regarding anger regulation, we referred to the taxonomy of adaptive and maladaptive anger regulation processes proposed by Spielberger and colleagues [59]. Among maladaptive processes, “anger out” refers to a failure in anger regulation that leads to excessive or inappropriate expressions of anger towards other persons or objects, while “anger in” refers to the tendency to hold in, turn to the self and suppress angry feelings. The adaptive form of regulation refers to active attempts to avoid anger externalization through physical and verbal expressions (“anger control out”) or by calming down or cooling off (“anger control in”). Our main hypothesis was that attachment insecurity is associated with greater use of maladaptive forms of emotion regulation and less anger control, and that these anger regulation difficulties, in turn, exacerbate BP features.

## 2. Method

### 2.1. Participants

Volunteers were recruited online via social media posts and snowball sampling to complete online electronic questionnaires. The questionnaires were prepared using Google Forms and disseminated through different social media (Facebook and WhatsApp). The Google Form link was initially shared on social media and participants were encouraged to pass it on to others, with a focus on recruiting the general public (snowball sampling). Inclusion criteria were: (a) age 18 and older; (b) Italian speakers; and (c) complete answers on all questionnaires (no missing data).

The required sample size was computed based on the results reported by Fritz and MacKinnon [60], who computed the necessary sample sizes for the most common and the most recommended tests of mediation for various combinations of parameters. To apply their estimates, we should anticipate the sizes of the associations between attachment orientations and anger regulation (*α*) and between anger regulation and BP features (*β*). A previous study by Scott et al. [53] showed that the standardized path from attachment anxiety to negative affect (which included anger) was 0.66, whereas the path from a negative effect to BP features was 0.68. However, the path from attachment avoidance to a negative effect was not significant (−0.03). We therefore took a conservative position, by anticipating that the to-be-estimated *α* and *β* paths were small (0.26) and medium (0.39), respectively. With these estimates, we needed 115 participants to have a 0.80 power to find significant mediational effects, using the bias-corrected bootstrap method (i.e., the method adopted in the present study). Thus, the final sample consisted of 302 adults, including 224 females (74.17%) and 77 males (25.50%)—one participant responded other (0.33%). Age ranged between 18 and 73 years (M = 37.16, SD = 11.71). Specifically, 11 participants were less than 20 years old (3.64%), 72 were between 21 and 30 years old (23.84%), 85 were between 31 and 40 years old (28.15%), 33 were between 41 and 50 years old (10.93%), 25 were between 51 and 60 years old (8.28%), and 14 were more than 60 years old (4.64%). Lastly, for education, 21 participants had a secondary school degree (6.95%), 122 had a high school degree (40.40%), 118 had a university degree (39.07%), and 41 had a post-graduate degree (13.58%). This study received approval from the Ethical Committee of the University of Trento (protocol 2019-035). Informed consent was obtained from all participants included in the study.

### 2.2. Instruments

Borderline features scale of the Personality Assessment Inventory (PAI-BOR, [61]). The PAI-BOR is a 24-item self-reporting measure that assesses features associated with BP (total score: *α* = 0.86). Four subscales of the PAI-BOR target Affective Instability (e.g., “My mood can shift quite suddenly”; *α* = 0.80), Identity Problems (e.g., “My attitude about myself changes a lot”; *α* = 0.63), Negative Relationships (e.g., “My relationships have been stormy”; *α* = 0.56), and Self-harm (e.g., “When I am upset, I typically do something to hurt myself”; *α* = 0.80). These items are rated on a four-point Likert scale (ranging from 0 = “false” to 3 = “very true”). The Italian version of the PAI-BOR has demonstrated reliability and validity in non-clinical samples [62].

Experiences in Close Relationships-12 (ECR-12; [63]). The ECR-12 is a 12-item self-reporting questionnaire used to examine attachment style, with items relating to perceptions and emotional experiences in romantic relationships. The ECR-12 is composed of two 6-item subscales: Attachment Anxiety (e.g., “*I worry that romantic partners won’t care about me as much as I care about them*”) and Attachment Avoidance (e.g., “*I don’t feel comfortable opening up to romantic partners*”), with higher scores indicating more anxious and avoidant attachment styles, respectively. The ECR-12 maintains strong psychometric properties despite its relatively abbreviated form (compared with previous iterations of the ECR: [64], with a robust fit of the two-dimensional structure and high internal consistency [63]. In the Italian version, Cronbach’s alphas were 0.85 for the Anxiety subscale and 0.86 for the Avoidance subscale [65].

STAXI-2. The State and Trait Anger Expression Inventory-2 (STAXI-2, [59]) is a 57-item questionnaire used to assess state anger, trait anger, anger control (inward and outward) and anger expression (inward and outward). State Anger (15 items, e.g., “I feel angry”) refers to the intensity of anger at the time of testing, while Trait Anger refers to the individual disposition to experience feelings of anger with variable intensity, frequency, and duration (10 items, e.g., “I am a hothead person”). Anger-Expression-In refers to the extent to which people hold things in or suppress their emotions (i.e., the internalization/suppression of anger; 8 items, e.g., “I keep things in”), while Anger-Expression-Out refers to the extent to which people express their anger outwardly in a poorly controlled manner (i.e., the externalization of anger; 8 items, e.g., “I do things like slam doors”). Finally, the Anger-Control-In and Anger-Control-Out subscales refer to people’s ability to monitor and control their emotions, by calming down (Anger-Control-In; 8 items, e.g., “I control my angry feelings”) and avoiding anger externalization through physical and verbal expressions (Anger-Control-Out; 8 items, e.g., “I control my behaviour”). In the Italian version, Cronbach’s alphas calculated for young adults and adults were acceptable, ranging between 0.73 and 0.88 [66].

### 2.3. Statistical Analyses

First, Pearson’s correlation coefficients were calculated to evaluate the correlations among the study variables considered in the present study—BPD symptoms in general and BPD symptomatic domains (Affective Instability, Identity Problems, Negative Relationships, and Self-harm) on the one hand, and attachment dimensions (Attachment Anxiety and Attachment Avoidance), Trait Anger, and anger expression and regulation dimensions (Anger-Expression-In, Anger-Expression-Out, Anger-Control-In, and Anger-Control-Out) on the other.

Second, following the scheme proposed by Frazier, Tix and Barron [67] to test mediational hypotheses, hierarchical regressions were conducted to examine (a) the associations between BPD symptoms and attachment styles (controlling for demographic variables), (b) the associations between attachment styles and the different aspects of anger regulation (controlling for demographic variables and Trait Anger), and (c) the simultaneous associations of anger regulation and attachment styles with BPD symptoms (controlling for demographic variables and Trait Anger).

Finally, we examined whether the four dimensions of anger regulation mediated the associations between attachment styles and BPD symptoms. For this purpose, we used the PROCESS macros (Model 4: [68]). The significance of the direct and indirect effects was examined via a bias-corrected bootstrapping procedure (5000 samples), using a 95% confidence interval [69]. The consensus is that if the confidence interval does not contain zero, then the indirect effect can be considered significant [68].

## 3. Results

### 3.1. Preliminary Analyses

Table 1 reports the descriptive statistics for the variables examined in the present study, both for the whole sample and separately for males and females. In all cases, the asymmetry and kurtosis values were in the ranges recommended by Hair et al. [70] to prove normal univariate distribution (skewness ranged from −0.36 to +1.41, whereas kurtosis ranged from −0.95 to +3.61), suggesting that parametric analyses could be applied. We conducted preliminary analyses (*t*-tests for independent samples) to ascertain potential differences related to gender. As can be seen in Table 1 a significant difference occurred only for the Anger-Control-In subscale of the STAXI (*t*(299) = 2.18, *p* = 0.030), with females reporting a better ability to control their emotions as compared to males. All other analyses were non-significant (*p* > 0.12).

We also investigated bivariate correlations with age and education. For age, the results (illustrated in Table 2) showed significant negative correlations with the Trait Anger and Anger-Expression-In subscales of the STAXI, the total scores of the PAI-BOR, and the Attachment Anxiety subscale of the ECR-12. Thus, when compared with younger adults, older adults were less likely to experience angry feelings, to suppress their emotions, to report BPD symptoms, and to exhibit anxious attachment styles. On the other hand, age was positively correlated with the Anger-Control-In and Anger-Control-Out subscales of the STAXI, suggesting that older adults were better able to control their emotions and behaviours.

Similar results were obtained for education (see Table 2). Specifically, this variable showed (a) negative correlations with the Trait Anger, Anger-Expression-In and Anger-Expression-Out subscales of the STAXI, and the total scores of the PAI-BOR, and (b) positive correlations with the Anger-Control-In and Anger-Control-Out subscales of the STAXI.

### 3.2. Bivariate Correlations

As illustrated in Table 2, BP features were significantly and positively correlated with both Attachment Anxiety and Attachment Avoidance, with a stronger contribution of Attachment Anxiety. Thus, participants who had high levels of attachment anxiety and avoidance were also more likely to report BP features. Considering the specific symptomatic dimensions, Attachment Anxiety was correlated with all symptomatic dimensions of BP (Affect Instability, Identity Problems, Negative Relationships, and Self-Harm), whereas Attachment Avoidance was correlated only with Affect Instability, Self-Harm and Negative Relationships.

Of interest for the present purposes, we also found that attachment anxiety was positively and significantly associated with the Trait Anger, Anger-Expression-In and Anger-Expression-Out subscales of the STAXI, and negatively associated with the Anger-Control-Out subscale. Thus, participants having high levels of attachment anxiety were more inclined to experience angry feelings, to suppress their emotions, and to express their anger through outward behaviours; in addition, they were less likely to control their behaviours. For attachment avoidance, the analyses revealed significant correlations with the Anger-Expression-In (positive) and Anger-Control-In (negative) subscales of the STAXI. This indicated that participants having high levels of attachment avoidance were less able to control their emotions and hence more likely to suppress them.

Finally, BP features in general and all specific BP feature dimensions were positively correlated with the Trait Anger, Anger-Expression-In and Anger-Expression-Out subscales of the STAXI, but negatively correlated with the Anger-Control-In and Anger-Control-Out subscales. Therefore, participants who were more inclined to experience angry feelings, to suppress their emotions, and to express their anger with outward behaviours reported more severe BP features. In contrast, participants who were more able to control their emotions and behaviours reported less severe BP features.

### 3.3. Regression Analyses

Hierarchical regression analyses were performed in three steps [67]. First, we determined whether attachment styles predicted BD features after removing the effects due to age, gender and education. Table 3 shows that the contributions of attachment anxiety and attachment avoidance were significant in all cases; thus, the to-be-mediated links between the predictors and the outcome measures were robust.

Second, the scores of the four anger regulation subscales were regressed in terms of attachment styles to establish whether the links between the predictors and the mediators were significant. As reported in Table 4, after controlling for age, gender, education and trait anger, it turned out that both attachment anxiety and avoidance were positively associated with the Anger-Expression-In scores, while attachment avoidance was negatively associated with the Anger-Control-In and Anger-Control-Out scores.

Third, both the predictors (attachment styles) and the mediators (anger regulation subscales) were regressed in terms of BDP symptoms. This allowed us to confirm that the mediators were related to the outcome, and to provide an estimate of the relation between the predictors and the outcome controlling for the mediators. Table 5 shows that the total BP features were positively related to both attachment anxiety and avoidance, and to Anger-Expression-In and Anger-Expression-Out scores, but negatively related to Anger-Control-Out scores. Importantly, when comparing the coefficient scores in Table 3 and Table 4, it can be seen that the *β* coefficients of attachment anxiety and avoidance were substantially reduced after the introduction of the anger regulation subscales (from 0.49 to 0.34 for attachment anxiety; from 0.26 to 0.16 for attachment avoidance). These findings are consistent with the idea that anger regulation skills partially mediate the links between attachment styles and BD. Comparable results were obtained for the specific BP; the contributions of attachment anxiety and attachment avoidance were significant but reduced in size—except for the Identity Problems subscale, for which the contribution of attachment avoidance was marginal (suggesting an almost full mediation). The inspection of Table 5 is also interesting because it indicates that the contributions of the anger regulation measures were not the same across the four BDP symptoms. Indeed, the pattern observed for the total BP (positive associations with Anger-Expression-In and Anger-Expression-Out and negative associations with Anger-Control-Out) was substantially maintained in the Affective Instability and Identity Problems subscales. In contrast, for the Negative Relationships and Self-Harm subscales, significant or marginally significant contributions were only provided by the Anger-Control-Out (for Self-Harm) and Anger-Expression-Out (for Negative Relationships) measures.

### 3.4. Mediation Analyses

To formally verify whether anger regulation mediated the associations between attachment styles and BDP symptoms, we used the PROCESS software [68]. Specifically, we selected Model 4, which allowed us to enter up to 10 mediators operating in parallel. Thus, by using this model, we were able to enter all four anger subscales together and determine the significance of their mediating roles. Demographic variables (age, gender and education) and trait anger were entered as covariates to remove their confounding effects. Table 6 and Table 7 report the standardized indirect effects. Starting from attachment anxiety (Table 6), the results showed that the total indirect effect was significant, as the confidence interval did not contain zero (*β* = 0.07, 95% CI: 0.02/0.12). Specifically, attachment anxiety had a positive indirect effect on total BDP symptoms by increasing Anger-Expression-In scores (*β* = 0.06; 95% CI: 0.02/0.10); thus, participants having high levels of attachment anxiety were more likely to hold in and suppress their emotions, and this increased the severity of their BD features (see Figure 1). With respect to the specific BD dimensions, Table 6 indicates that the mediation was significant for the Affective Instability, Identity problems and Negative Relationships subscales, but not for the Self-Harm subscale.

A similar pattern of results was obtained for attachment avoidance (Table 7). The total indirect effect was significant (*β* = 0.10, 95% CI: 0.06/0.16). Attachment avoidance had a positive indirect effect on total BP by enhancing Anger-Expression-In scores (*β* = 0.07; 95% CI: 0.03/0.12); hence, even in this case, participants having high levels of attachment avoidance were more likely to hold in and suppress their emotions, and this resulted in more BP features (see Figure 2). When considering the specific BP, the mediation analyses were again significant for the Affective Instability, Identity problems and Negative Relationships subscales, but not for the Self-Harm subscale. In addition, it is worth noting that the mediation of Anger-Control-Out was marginally significant for the total BP scores, as well as for the Affective Instability and Identity problems subscales. Thus, participants having high levels of attachment avoidance tended to be less efficient in controlling their behaviours, and this resulted in more BP scores.

## 4. Discussion

Anger regulation is a core aspect of BP, and attachment orientations strongly influence the way individuals manage their emotions. This study was designed to investigate the role of different aspects of anger regulation as potential mediators of the associations between attachment orientations and several dimensions of BP features, including affect instability, identity problems, relational difficulties and self-harm.

The analysis of the associations between attachment orientations and BP confirmed previous evidence indicating that attachment anxiety and avoidance are strongly associated with BP features in both non-clinical [32,71]) and clinical populations [72,73]. Of note, we observed such associations considering both the total BP scores and the four symptomatic domains tapped by the PAI-BOR questionnaire (Affective Instability, Identity Problems, Negative Relationships, and Self-Harm).

With regard to the associations between anger regulation and attachment, we identified the internal expression of anger as the most relevant source of individual differences associated with attachment. This subscale refers to the extent to which people hold things in and suppress anger when they are angry [59]. These variables were positively associated with both anxious and avoidant attachment orientations, and were also predictive of BP. Importantly, the mediation analyses showed that the suppression of anger was also a significant mediator in the association between insecure attachment orientations and BP. Thus, participants having high levels of attachment insecurity were more likely to hold in their angry reactions, and this tendency, in turn, was predictive of higher scores in BP.

On the basis of the previous literature on attachment-related differences in emotion regulation [45,46,48], the association between the tendency to suppress anger and attachment avoidance is expected, while in the case of attachment anxiety, the expectation is an association with the tendency to under-regulate emotions [74] Instead, our results are more in line with the hypothesis that attachment-related differences in emotion regulation are emotion-specific [50,51]. According to this hypothesis, attachment-related differences in emotion regulation are oriented to the satisfaction of attachment needs. Individuals who are high in attachment anxiety tend to under-regulate other negative emotions, but they may instead implement a suppression strategy when dealing with anger specifically [52]. Suppressing anger, indeed, may be more congruent with the attachment goal of abandonment avoidance [75]. Even though the literature concerning the specific dynamics of anger regulation in association with attachment styles is still scarce, the association between attachment anxiety and anger suppression has indeed been reported in previous studies [52,76].

The specificity of attachment-related differences in anger regulation may also explain the results concerning the external control of anger. The external control of anger is defined as the expenditure of energy to monitor and control the physical or verbal expressions of outward anger (for example, they may increase the risk of being aggressive or hostile toward other people or objects) [5]. This variable was negatively associated with BP avoidant attachment orientation (whereas the association with attachment anxiety was not significant). In this case, the mediational analysis showed that the external control of anger was a marginal mediator in the association between avoidant attachment orientation and BP (especially for the Affective Instability and Identity Problems subscales). Thus, participants having high levels of attachment avoidance were less likely to externally control anger, and this tendency, in turn, was predictive of higher scores of BP. The interaction of high suppression and low external control of anger may reflect the dynamics of maladaptive anger rumination, which was previously reported as a characteristic of BP in both clinical [29,77,78] and non-clinical populations [78]. According to the “emotional cascade model” [79], high levels of negative effects in people with BP trigger rumination, which in turn intensifies the negative effect. As part of a vicious cycle, the intensification of the negative effect may lead to the dysregulated behaviours described in BP patients, such as self-harm or substance abuse [25,28,30].

The lack of behavioural control is in line with the evidence of aggressive and self-harm behaviours in BP patients [80], suggesting that difficulties in emotional control could be a critical factor in contributing to the behavioural characteristics of BP disorder. Of note, in the present study, BP features in the domain of self-harm were predicted by less control of outside anger, whereas it was not significantly predicted by suppression. Thus, the poor capacity to control anger in particular could determine a higher rate of aggressive and self-harm behaviours, possibly suggesting that the incapacity to control anger spreads both outwardly and inwardly. In this sense, rumination and aggressive behaviour can be viewed as two faces of the same coin of emotion regulation difficulties, leading to cognitive or behavioural avoidance of adaptive anger and increased anger dysregulation [81].

There are a few noteworthy limitations in the present study. First, although our sample was reasonably large, it was composed of healthy participants. In future studies, working on samples composed of individuals affected by borderline personality disorder may help to better establish the relationship between borderline personality, attachment orientations and anger regulation. Second, this study relied entirely on self-reported measures. Future studies will likely benefit from the use of more sophisticated diagnostic procedures, such as clinician-administered diagnostic measures for the assessment of borderline personality, and semi-structured interviews for the assessment of attachment styles (e.g., Adult Attachment Interview; [82]). Moreover, among self-reports, the subscale Negative Relationships of the PAI-BOR has limited reliability (*α* = 0.56); thus, results concerning this domain must be considered with hesitation. Finally, although the mediation models involving anger regulation were statistically significant, the cross-sectional design of our study makes it impossible to draw causal inferences.

### Clinical Implications

Attachment-related behaviours are theorized to be stable from childhood to adulthood [83] and this hypothesis has been largely confirmed by attachment research [84,85]. Even if research supports the suggestion that attachment styles may change during the course of psychotherapy ([86,87], this achievement can be considered difficult. Due to this difficulty in changing attachment styles, the examination of the mediating effects of other variables, which may be better modified through clinical interventions, has relevant clinical implications. In recent years, emotion regulation strategies have emerged as a key element for the reduction of psychological difficulties associated with attachment [88]. In this perspective, the results of the present study may have crucial clinical implications for the treatment of BPD.

First, we suggest that psychological treatments of BP patients should aim for the reduction of the “holding in” strategy to regulate anger. Among others, useful therapeutic techniques may include encouraging the identification and expression of anger (expressive techniques), as well as exposure to anger (behavioural techniques) and the parallel building of anger tolerance through the promotion of acceptance-based regulation, as opposed to suppression and rumination [89]. Considering the observed interplay between attachment and anger regulation, the therapeutic relationship may also have a key role. The adaptive expression of anger can take advantage of the security of the therapeutic setting. Moreover, encouraging and validating adaptive anger expression, as well as expressing empathy, would help in the construction of more adaptive ways of expressing spontaneous emotions [90,91]. Second, when ruminative processes or uncontrolled behavioural expressions of anger are active, therapeutic interventions can be used to block emotion dysregulation and promote the use of more adaptive strategies of emotion regulation [92,93,94,95]. For example, useful techniques may include the identification of maladaptive forms of regulation, distinguishing adaptive anger expression from suppression or rumination, paying attention to the negative consequences of turning anger toward the self (anger in), and all forms of anger mentalization techniques.

## Figures and Tables

**Figure 1 brainsci-13-00878-f001:**
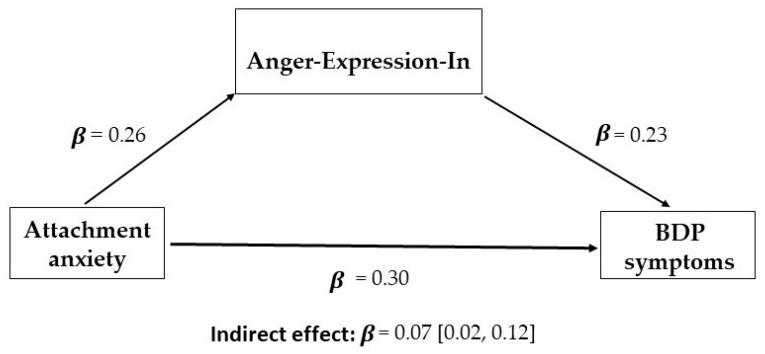
Mediation model for the indirect effect of attachment anxiety on BDP symptoms.

**Figure 2 brainsci-13-00878-f002:**
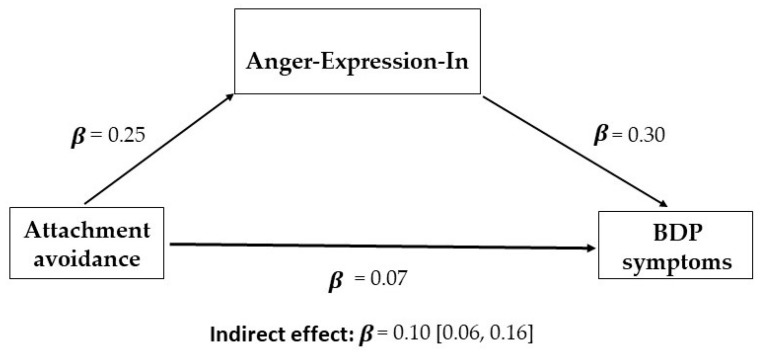
Mediation model for the indirect effect of attachment avoidance on BDP symptoms.

**Table 1 brainsci-13-00878-t001:** Descriptive statistics for the variables measured in the present study and gender differences.

Variable	Total Sample	Females	Males	*t*-Test
ECR-12				
Attachment anxiety	24.56 (9.56)	24.05 (9.63)	25.95 (9.33)	*t*_(299)_ = −1.50
Attachment avoidance	13.97 (7.70)	13.99 (7.82)	13.84 (7.39)	*t*_(299)_ = 0.14
PAI-BOR				
Affective instability	6.66 (3.89)	6.74 (3.75)	6.42 (4.32)	*t*_(299)_ = 0.63
Identity problems	8.18 (4.27)	8.40 (4.34)	7.53 (4.03)	*t*_(299)_ = 1.53
Negative relationships	7.89 (3.44)	7.97 (3.30)	7.66 (3.84)	*t*_(299)_ = 0.67
Self-harm	3.62 (3.47)	3.52 (3.35)	3.92 (3.83)	*t*_(299)_ = −0.87
Total scores	26.35 (12.11)	26.63 (11.65)	25.53 (13.48)	*t*_(299)_ = 0.68
STAXI				
Trait anger	18.89 (4.66)	18.95 (4.43)	18.74 (5.32)	*t*_(299)_ = 0.33
Anger-Expression-In	17.80 (4.67)	17.81 (4.57)	17.73 (4.97)	*t*_(299)_ = 0.13
Anger-Expression-Out	13.30 (3.33)	13.25 (3.18)	13.52 (3.77)	*t*_(299)_ = −0.62
Anger-Control-In	21.64 (4.50)	21.97 (4.57)	20.68 (4.22)	*t*_(299)_ = 2.18 *
Anger-Control-Out	24.17 (4.77)	24.40 (4.65)	23.45 (5.09)	*t*_(299)_ = 1.50

Note. *: *p* < 0.05.

**Table 2 brainsci-13-00878-t002:** Pearson’s correlations between all variables. Correlations reported in bold were significant (*p* < 0.05), while correlations reported in italic were marginally significant (0.05 < *p* < 0.06).

Variables	1	2	3	4	5	6	7	8	9	10	11	12	13	14
1. Age	1.00													
2. Education	0.09	1.00												
3. Attachment anxiety	−0.22	−0.08	1.00											
4. Attachment avoidance	0.06	−0.07	−0.13	1.00										
5. Trait anger	−0.20	−0.15	0.21	0.03	1.00									
6. Anger-Expression-In	−0.25	−0.12	0.33	0.25	0.24	1.00								
7. Anger-Expression-Out	−0.09	−0.19	0.11	0.06	0.69	*0.11*	1.00							
8. Anger-Control-In	0.14	0.15	*−0.10*	−0.15	−0.31	−0.01	−0.32	1.00						
9. Anger-Control-Out	0.19	0.16	−0.19	*−0.11*	−0.54	−0.03	−0.60	0.66	1.00					
10. BDP symptoms (Total)	−0.30	−0.22	0.50	0.20	0.57	0.44	0.49	−0.33	−0.50	1.00				
11. Affective instability	−0.28	−0.22	0.40	0.22	0.55	0.43	0.48	−0.35	−0.51	0.88	1.00			
12. Identity Problems	−0.39	−0.20	0.58	*0.10*	0.44	0.49	0.34	−0.28	−0.42	0.85	0.71	1.00		
13. Negative Relationships	−0.18	−0.14	0.38	0.13	0.40	0.28	0.34	−0.18	−0.28	0.77	0.58	0.55	1.00	
14. Self-Harm	*−0.09*	−0.15	0.21	0.18	0.43	0.16	0.39	−0.22	−0.39	0.68	0.49	0.39	0.35	1.00

**Table 3 brainsci-13-00878-t003:** Hierarchical regressions predicting BDP symptoms from attachment styles.

Predicted Measure		Predictors	*β*	t	∆R^2^	F Change
BPD Symptoms	Step 1	Age	−0.20	−4.39 **	0.14	*F* = 16.06 **
		Gender	0.09	2.02 *		
		Education	−0.15	−3.49 **		
	Step 2	Attachment anxiety	0.49	10.49 **	0.26	*F* = 65.44 **
		Attachment avoidance	0.26	5.85 **		
Affective instability	Step 1	Age	−0.20	−4.05 **	0.12	*F* = 14.14 **
		Gender	0.08	1.64		
		Education	−0.16	−3.38 **		
	Step 2	Attachment anxiety	0.38	7.69 **	0.18	*F* = 40.33 **
		Attachment avoidance	0.27	5.57 **		
Identity problems	Step 1	Age	−0.27	−6.35 **	0.19	*F* = 23.70 **
		Gender	0.14	3.44 **		
		Education	−0.12	−2.99 **		
	Step 2	Attachment anxiety	0.54	12.52 **	0.28	*F* = 82.54 **
		Attachment avoidance	0.18	4.41 **		
Negative relationships	Step 1	Age	−0.09	−1.82 †	0.05	*F* = 5.35 **
		Gender	0.07	1.49		
		Education	−0.09	−1.84 †		
	Step 2	Attachment anxiety	0.39	7.28 **	0.16	*F* = 30.24 **
		Attachment avoidance	0.18	3.61 **		
Self-Harm	Step 1	Age	−0.05	−0.88	0.03	*F* = 3.31 *
		Gender	−0.02	−0.47		
		Education	−0.11	−2.10 *		
	Step 2	Attachment anxiety	0.21	3.81 **	0.07	*F* = 12.52 **
		Attachment avoidance	0.20	3.70 **		

Note. †: 0.05 < *p* < 0.10; *: *p* ≤ 0.05; **: *p* ≤ 0.01.

**Table 4 brainsci-13-00878-t004:** Hierarchical regressions predicting anger regulation subscales from attachment styles.

Predicted Measure		Predictors	*β*	t	∆R^2^	F Change
Anger-Expression-In	Step 1	Age	−0.18	−3.48 **	0.07	*F* = 8.27 **
		Gender	0.03	0.67		
		Education	−0.04	−0.77		
	Step 2	Trait Anger	0.12	2.29 *	0.18	*F* = 23.66 **
		Attachment anxiety	0.30	5.73 **		
		Attachment avoidance	0.29	5.72 **		
Anger-Expression-Out	Step 1	Age	0.05	1.23	0.04	*F* = 4.48 **
		Gender	−0.04	−1.15		
		Education	−0.08	−2.11 *		
	Step 2	Trait Anger	0.69	16.10 **	0.45	*F* = 88.99 **
		Attachment anxiety	−0.03	−0.68		
		Attachment avoidance	0.03	0.72		
Anger-Control-In	Step 1	Age	0.07	1.34	0.05	*F* = 5.58 **
		Gender	0.12	2.26 *		
		Education	0.08	1.57		
	Step 2	Trait Anger	−0.27	−4.82 **	0.09	*F* = 11.28 **
		Attachment anxiety	−0.03	−0.63		
		Attachment avoidance	−0.14	−2.68 **		
Anger-Control-Out	Step 1	Age	0.06	1.39	0.06	*F* = 6.62 **
		Gender	0.08	1.80		
		Education	0.05	1.22		
	Step 2	Trait Anger	−0.50	−10.08 **	0.27	*F* = 39.66 **
		Attachment anxiety	−0.09	−1.96 *		
		Attachment avoidance	−0.11	−2.49 **		

Note. *: *p* < 0.05; **: *p* < 0.01.

**Table 5 brainsci-13-00878-t005:** Hierarchical regressions predicting BDP symptoms from attachment styles and anger regulation subscales.

Predicted Measure		Predictors	*β*	t	∆R^2^	F Change
BPD Symptoms	Step 1	Age	−0.09	−2.47 **	0.14	*F* = 16.06 **
		Gender	0.09	2.49 **		
		Education	−0.07	−2.03 *		
	Step 2	Trait Anger	0.21	4.00 **	0.48	*F* = 53.52 **
		Anger-Expression-In	0.17	4.17 **		
		Anger-Expression-Out	0.13	2.44 *		
		Anger-Control-In	−0.02	−0.42		
		Anger-Control-Out	−0.18	−3.02 **		
		Attachment anxiety	0.34	8.51 **		
		Attachment avoidance	0.16	4.07 **		
Affective instability	Step 1	Age	−0.08	−1.91 †	0.12	*F* = 14.14 **
		Gender	0.08	2.09 *		
		Education	−0.07	−1.90 †		
	Step 2	Trait Anger	0.19	3.39 **	0.43	*F* = 40.32 **
		Anger-Expression-In	0.22	4.74 **		
		Anger-Expression-Out	0.11	1.97 *		
		Anger-Control-In	−0.04	−0.81		
		Anger-Control-Out	−0.21	−3.37 **		
		Attachment anxiety	0.22	5.08 **		
		Attachment avoidance	0.14	3.43 **		
Identity problems	Step 1	Age	−0.17	−4.42 **	0.19	*F* = 23.70 **
		Gender	0.14	3.99 **		
		Education	−0.06	−1.71 †		
	Step 2	Trait Anger	0.08	1.55	0.41	*F* = 44.38 **
		Anger-Expression-In	0.25	5.72 **		
		Anger-Expression-Out	0.06	1.23		
		Anger-Control-In	−0.04	−0.84		
		Anger-Control-Out	−0.17	−2.89 **		
		Attachment anxiety	0.40	9.83 **		
		Attachment avoidance	0.07	1.76 †		
Negative relationships	Step 1	Age	−0.04	−0.83	0.05	*F* = 5.35 **
		Gender	0.07	1.44		
		Education	−0.04	−0.85		
	Step 2	Trait Anger	0.19	2.68 **	0.25	*F* = 15.37 **
		Anger-Expression-In	0.06	1.08		
		Anger-Expression-Out	0.14	1.91 †		
		Anger-Control-In	−0.02	−0.43		
		Anger-Control-Out	0.01	−0.19		
		Attachment anxiety	0.32	5.79 **		
		Attachment avoidance	0.14	2.68 **		
Self-Harm	Step 1	Age	0.01	0.25	0.03	*F* = 3.31 *
		Gender	−0.02	−0.57		
		Education	−0.05	−1.05		
	Step 2	Trait Anger	0.22	3.05 **	0.23	*F* = 13.47 **
		Anger-Expression-In	0.00	0.10		
		Anger-Expression-Out	0.11	1.44		
		Anger-Control-In	0.05	0.82		
		Anger-Control-Out	−0.18	−2.19 *		
		Attachment anxiety	0.13	2.41 *		
		Attachment avoidance	0.16	3.00 **		

Note. †: 0.05 < *p* < 0.10; *: *p* < 0.05; **: *p* < 0.01.

**Table 6 brainsci-13-00878-t006:** Standardized coefficients for the indirect effects of attachment anxiety on BDP symptoms. The coefficients highlighted in bold were significant.

Predicted Measure	Mediators	*β*	95% CI	z-Value
BDP Symptoms (Total)	Total indirect effect	0.070	0.022; 0.123	z = 3.27 **
	Anger-Expression-Out	−0.004	−0.022; 0.007	z = −0.76
	Anger-Expression-In	0.063	0.028; 0.105	z = 3.72 **
	Anger-Control-Out	0.011	−0.008; 0.038	z = 1.12
	Anger-Control-In	0.001	−0.006; 0.009	z = 0.26
Affective instability	Total indirect effect	0.083	0.036; 0.131	z = 3.44 **
	Anger-Expression-Out	−0.004	−0.014; 0.006	z = −0.74
	Anger-Expression-In	0.072	0.035; 0.110	z = 3.82 **
	Anger-Control-Out	0.014	−0.010; 0.038	z = 1.13
	Anger-Control-In	0.001	−0.006; 0.008	z = 0.26
Identity problems	Total indirect effect	0.082	0.039; 0.124	z = 3.81 **
	Anger-Expression-Out	−0.002	−0.009; 0.004	z = −0.67
	Anger-Expression-In	0.072	0.036; 0.109	z = 3.92 **
	Anger-Control-Out	0.010	−0.008; 0.030	z = 1.11
	Anger-Control-In	0.001	−0.005; 0.006	z = 0.26
Negative relationships	Total indirect effect	0.026	−0.009; 0.062	z = 1.45
	Anger-Expression-Out	−0.004	−0.018; 0.008	z = −0.74
	Anger-Expression-In	0.030	0.000; 0.061	z = 1.97 *
	Anger-Control-Out	0.000	−0.009; 0.009	z = 0.02
	Anger-Control-In	0.001	−0.004; 0.006	z = 0.25
Self-Harm	Total indirect effect	0.025	−0.013; 0.065	z = 1.27
	Anger-Expression-Out	−0.003	−0.014; 0.006	z = −0.70
	Anger-Expression-In	0.017	−0.011; 0.047	z = 1.18
	Anger-Control-Out	0.012	−0.009; 0.033	z = 1.07
	Anger-Control-In	−0.001	−0.004; 0.003	z = −0.23

Note. *: *p* < 0.05; **: *p* < 0.01.

**Table 7 brainsci-13-00878-t007:** Standardized coefficients for the indirect effects of attachment avoidance on BDP symptoms. The coefficients highlighted in bold were significant.

Predicted Measure	Mediators	*β*	95% CI	z-Value
BDP Symptoms	Total indirect effect	0.105	0.058; 0.152	z = 4.43 **
(Total)	Anger-Expression-Out	0.003	−0.005; 0.012	z = 0.74
	Anger-Expression-In	0.076	0.038; 0.114	z = 3.92 **
	Anger-Control-Out	0.023	−0.002; 0.049	z = 1.74 †
	Anger-Control-In	0.002	−0.012; 0.017	z = 0.29
Affective instability	Total indirect effect	0.109	0.061; 0.157	z = 4.47 **
	Anger-Expression-Out	0.003	−0.005; 0.011	z = 0.73
	Anger-Expression-In	0.075	0.037; 0.114	z = 3.89 **
	Anger-Control-Out	0.024	−0.002; 0.052	z = 1.76 †
	Anger-Control-In	0.005	−0.010; 0.021	z = 0.70
Identity problems	Total indirect effect	0.130	0.077; 0.183	z = 4.81 **
	Anger-Expression-Out	0.001	−0.003; 0.005	z = 0.40
	Anger-Expression-In	0.099	0.052; 0.146	z = 4.17 **
	Anger-Control-Out	0.024	−0.003; 0.051	z = 1.74 †
	Anger-Control-In	0.005	−0.011; 0.021	z = 0.61
Negative relationships	Total indirect effect	0.056	0.016; 0.096	z = 2.77 **
	Anger-Expression-Out	0.003	−0.006; 0.014	z = 0.72
	Anger-Expression-In	0.044	0.011; 0.078	z = 2.61 **
	Anger-Control-Out	0.004	−0.011; 0.020	z = 0.60
	Anger-Control-In	0.003	−0.015; 0.022	z = 0.34
Self-Harm	Total indirect effect	0.028	−0.011; 0.068	z = 1.40
	Anger-Expression-Out	0.003	−0.005; 0.012	z = 0.69
	Anger-Expression-In	0.013	−0.014; 0.041	z = 0.96
	Anger-Control-Out	0.019	−0.005; 0.044	z = 1.56
	Anger-Control-In	−0.008	−0.028; 0.011	z = −0.81

Note. †: *p* < 0.1; **: *p* < 0.01.

## Data Availability

Data available on request from the authors.

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
