# Peer review of "“Holding in Anger” as a Mediator in the Relationship between Attachment Orientations and Borderline Personality Features"

_brainsci, 2023, doi:10.3390/brainsci13060878_

Round 1
Reviewer 1 Report
Overall, by studying the roles of attachment and anger management in a subclinical population, this study adds significantly to the body of knowledge on borderline personality (BP) and emotional dysfunction. The 302 participant sample size is sufficient for the analyses performed, and the study's research objectives and methodology are well laid out.
The study's findings are consistent with the hypothesis that anger regulation, specifically the suppression of anger, mediates the influence of attachment on BP characteristics. According to this result, those who have insecure attachment styles are more prone to repress their anger, which may in turn help people develop BP traits. Additionally, the authors point out that anger control may serve as a weakly significant partial mediator in the association between attachment avoidance and BP features.
Comments
Table 1 - Incomplete where is the Mean/SD
Table 5 its unusual to use / to separate the 95%CI please use - or ;
Clarify limitations
Add doi to refs
Author Response
Dear Reviewer,
thank you for reviewing our manuscript.
- We had another table in the appendix with Ms and SDs. We moved this table in the manuscript as Table 1 (and we changed therefore the numbers of the other tables)
- we corrected the Table 5 (now Table 6)
- We added a description of the limitations
- We added the DOIs to the references
Reviewer 2 Report
“Holding in Anger” as a Mediator in The Relationship Between Attachment Orientations and Borderline Personality Features
This is an interesting and important contribution to the mental health field. The article is well written and methodologically sound. Still, I believe a few minor changes would improve the overall quality of the paper.
1. Please provide more specific information on how the sample was collected and how sample size was calculated.
2. The negative relationships subscale presents an α = .56, which is very concerning. This should be addressed as a limitation or results should not include this subscale.
3. What is the cronbach’s alpha for STAXI-2?
4. Please provide a limitations section.
Best wishes.
Author Response
Dear Reviewer,
Thank you very much for reviewing our manuscript.
According to the comments:
- we provide more information on data collection procedure and sample size calculation;
- we mentioned the limited alpha of the Negative Relationships sub-scale in the new section 'Limitation'
- we added the Cronbach's Alpha for the STAXI-2
- we provided a new section on limitation